# Efficacy of AutoPulse for Mechanical Chest Compression in Patients with Shock-Resistant Ventricular Fibrillation

**DOI:** 10.3390/ijerph19052557

**Published:** 2022-02-23

**Authors:** Jarosław Gorący, Paweł Stachowiak, Arkadiusz Krejczy, Patrycja Piątek, Iwona Gorący

**Affiliations:** 1Independent Laboratory of Invasive Cardiology, Pomeranian Medical University in Szczecin, 70-204 Szczecin, Poland; jargo@pum.edu.pl; 2Department of Cardiology, The Regional Specialist Hospital in Wrocław, Research and Development Center, 51-124 Wrocław, Poland; stachowiak22@gmail.com; 3Province Emergency Medical Services, 71-011 Szczecin, Poland; akrejczy@wp.pl; 4Intensive Care Unit for Children, Pomeranian Medical University, 71-252 Szczecin, Poland; 5Helicopter Emergency Medical Service, 01-934 Warszawa, Poland; 6Department of Cardiology, Pomeranian Medical University, 70-111 Szczecin, Poland; 7Doctoral Studies, Pomeranian Medical University, 71-210 Szczecin, Poland; 8Department of Clinical and Molecular Biochemistry, Pomeranian Medical University, 70-111 Szczecin, Poland; iwona.goracy@pum.edu.pl

**Keywords:** cardiac arrest, catheterization laboratory, mechanical chest compression, PCI, ventricular fibrillation

## Abstract

Introduction: Sudden cardiac arrest is one of the most common causes of death. In cases of shock-resistant ventricular fibrillation, immediate transport of patients to the hospital is essential and made possible with use of devices for mechanical chest compression. Objectives: The efficacy of AutoPulse in patients with shock-resistant ventricular fibrillation was studied. Methods: This is a multicentre observational study on a population of 480,000, with 192 reported cases of out-of-hospital cardiac arrest. The study included patients with shock-resistant ventricular fibrillation defined as cardiac arrest secondary to ventricular fibrillation requiring ≥3 consecutive shocks. Eventually, 18 patients met the study criteria. Results: The mean duration of resuscitation was 48.4±43 min, 55% of patients were handed over to the laboratory while still in cardiac arrest, 83.3% of them underwent angiography and, in 93.3% of them, infarction was confirmed. Coronary intervention was continued during mechanical resuscitation in 50.0% of patients, 60% of patients survived the procedure, and 27.8% of the patients survived. Conclusions: Resistant ventricular fibrillation suggests high likelihood of a coronary component to the cardiac arrest. AutoPulse is helpful in conducting resuscitation, allowing the time to arrival at hospital to be reduced.

## 1. Introduction

Sudden cardiac arrest (SCA) is one of the most common causes of death, with only 2–9% of SCA patients surviving until being discharged from the hospital [1]. Effective basic life support (BLS) provided until the arrival of a qualified ambulance team at the scene and further advanced life support (ALS) are the most important factors that increase the survival of patients with out-of-hospital cardiac arrest (OHCA) [2]. Ventricular fibrillation (VF) is one of the most common rhythms identified on the preliminary electrocardiogram (ECG) during a cardiac arrest [3]. Shock-resistant VF, on the other hand, is much less common, especially since the advent of biphasic defibrillation. If, however, the patient is found to have this arrhythmia, their chances of survival are even smaller, as the mortality of patients requiring prolonged cardiopulmonary resuscitation (CPR) is very high, reaching up to 95% [4]. Mechanical chest compression devices are useful in cases of prolonged cardiac arrest [4]. Thanks to these devices, patients with OHCA refractory to standard ALS can be transported directly to the hospital while CRP is being administered. It should be borne in mind that VF is identified in about 5% of patients with ST-segment elevation myocardial infarction [5], which is why it is of key importance that these patients are immediately taken to a hospital with a catheterization laboratory and undergo coronary angiography and percutaneous coronary intervention (PCI) [6]. 

The aim of the present analysis was to report on the efficacy of the AutoPulse (AP) system for mechanical chest compression (model 100 AutoPulse^®^ resuscitation system) in patients with shock-resistant VF transported directly to a catheterization laboratory during CPR. The AP platform comprised a board on which the patient was positioned and the LifeBand. A stroke of the patient’s heart was generated by mechanical shortening of the band, which caused the anteroposterior dimension of the chest to decrease by 20%. The device provided continuous chest compressions and worked in the continuous compression mode with asynchronous ventilation, as none of the patients had any ventilatory problems. 

## 2. Methods

This was a one-year prospective multicentre observational study conducted in an area inhabited by a population of 480 thousand people. All the hospitals with a catheterization laboratory providing 24/7 service and all the ambulance stations within the area of interest took part in the study. Before the study, all the ambulance teams within the region and all the catheterization laboratories were provided with AP systems. The effectiveness of only the AutoPuls (not other devices) was discussed, because in our region only this kind of device was more available. A total of 73 ambulance teams in the region were provided with these devices and 16 of these teams participated in the study within the area inhabited by the study population. Within this area, 192 cases of OHCA were reported during the one-year follow-up. Patients with shock-resistant VF were included in the study. Resistant VF was defined as a cardiac arrest secondary to VF that required a minimum of 3 consecutive shocks. The eligibility criteria were as follows: BLS provided prior to the arrival of an ambulance team or a cardiac arrest ensuing in the presence of the teamEndotracheal intubation or laryngeal mask airway (LMA) insertionAt least 3 consecutive episodes of VT in consecutive CPR cycles administered in accordance with the ERC guidelinesHigh likelihood of an acute coronary syndrome (ACS)The patient had to reach the catheterization laboratory in cardiac arrest or after return of spontaneous circulation (ROSC) and be considered alive.The exclusion criteria were as follows:Low values of end-tidal carbon dioxide (EtCO_2_)No possibility of providing mechanical chest compressionsNo possibility of rapid (up to 5 min) evacuation of the patient from the sceneLow likelihood of a coronary cause of the SCAPatient pronounced dead prior to reaching the catheterization laboratory.

The study endpoints included 30-day survival and discharge in a good neurological condition, and short-term survival, namely 24-h post-PCI survival. Eventually, 18 patients met the study eligibility criteria and none of the exclusion criteria.

Equipment and procedure: During the project, mechanical chest compression was provided with the AP-model 100 AutoPulse^®^ resuscitation system (ZOLL Medical Corporation, Chelmsford, MA, USA). The device enabled continuous resuscitation of the patient during transport. The AP platform comprised a board on which the patient was positioned and the LifeBand. A stroke of the patient’s heart was generated by mechanical shortening of the band, which caused the anteroposterior dimension of the chest to decrease by 20%. Patient parameters (adults weighing 136 kg or less, with a chest circumference of 76 to 130 cm and a chest width of 25 to 38 cm) and operating parameters of the AutoPulse system were compatible. The device provided continuous chest compressions and worked in the continuous compression mode without breaks for ventilation, as none of the patients had any ventilatory problems. Each patient had a laryngeal mask airway (LMA) or endotracheal tube inserted. 

Defibrillators: ZOLL E Series and X Series defibrillators (Chelmsford, MA, USA) were used by the ambulance teams. The protocol for defibrillation energy values for both defibrillator types is the same: 120 J for the first shock, 150 J for the second shock and 200 J for the third and subsequent shocks. 

The data were collected from the Independent Laboratory at the Invasive Cardiology Pomeranian Medical University, the Cardiology Department of Province Hospital in Stettin and the Province Emergency Medical Services in Stettin. A few of the researchers (Authors) participated directly in the study e.g., some were members of the emergency teams and interventional cardiology laboratory, while others were passive observers.

Deploying the system and staff qualifications: Upon the initial contact of the ambulance team with the patient, the procedure for using the AP system mandated that the paramedics take the system with them immediately to the patient. The AP system was deployed immediately after cardiac arrest confirmation. According to the procedure, upon delivering 3 shocks to the patient, a member of the ambulance team contacted the on-call physician at the catheterization laboratory and established with them whether the patient qualified for transport to the laboratory during provision of mechanical chest compressions. The staff attended periodic training in ALS, including the use of the device for mechanical chest compression and supraglottic airway devices. The coronary angiography and percutaneous coronary intervention (PCI) procedures were performed directly after the patient’s admission to the catheterization laboratory. The team performing the procedures always included two trained and experienced physicians, a technician and at least two nurses. The entire staff was well trained and experienced in administering CPR, which included the use of devices for mechanical chest compression. 

### 2.1. Ethics

The study was conducted according to the guidelines of the Declaration of Helsinki and approved by the Ethics Committee of the Pomeranian Medical University in Szczecin (protocol code KB-0012/35/03/2021/Z, date of approval: 8 March 2021). There was human participant involvement in this study. Patient consent was withdrawn due to the loss of consciousness resulting from sudden cardiac arrest. The patient had cardiac arrest and then cardiopulmonary resuscitation was performed using the AutoPulse device (patients with shock-resistant VT defined as a minimum of 3 consecutive shocks). The purpose of this activity was to save life. The use of a mechanical chest compression device (including AutoPulse)—when such a device is available—is a possible and correct procedure. The participant lost consciousness before inclusion in the study and was in a life-threatening condition (cardiac arrest), and written consent from the next of kin, who is the participant’s closest relative/guardian/legal authorized representative, was impossible to obtain. The time required to obtain such consent from a third party would be to the patient’s disadvantage. Activities were focused on saving life using an acceptable and available procedure. The study was approved by the Ethics Committee of the Pomeranian Medical University in Szczecin (protocol code KB-0012/35/03/2021/Z, date of approval: 8 March 2021). Informed consent was waived, and this was included in the approval of the study by the Ethics Committee. In Poland, when acceptable and available life-saving procedures are used, the consent of the patient or his relatives is not required (according to regulations of Polish law).

### 2.2. Statistical Analysis

The study variables are expressed as means and standard deviations (SD) or medians and interquartile ranges (IQR), as appropriate. In order to compare the patients who had developed the endpoint with those who had not, the Mann–Whitney U test was used for quantitative variables and the chi-square test with Fisher’s correction for nominal values was used for qualitative variables. Statistical calculations were performed using Statistica 12 (StatSoft Inc., Tulsa, OK, USA). A *p*-value < 0.05 was considered statistically significant.

## 3. Results

There were 18 patients included in the project of prospective evaluation of patient transport during cardiac arrest while performing CPR using the AP device (patients with shock-resistant VT, defined as a minimum of three consecutive shocks). The patients were aged from 43 to 80 years. Most of the patients (77.8%) were male. Most cases of SCA occurred in a public place (58.8% vs. 41.2% at home). An overwhelming majority of the cases occurred in the city (94.4% vs. 5.6% in the country). Table 1 provides the detailed study group characteristics along with the initial laboratory test results. The mean response time of the ambulance team was 4.6 ± 2min and the mean duration of CPR was 48.4 ± 43 min (range, 5–125 min). ROSC during transport to the hospital was achieved in 44.5% of the patients, while 55.5% of the patients were handed over to the catheterization laboratory while still in cardiac arrest. The mean duration of treatment provided by the ambulance team from the moment of CPR takeover or initiation to the moment of patient handover to the catheterization laboratory was 60.7 ± 16 min (median, 69 ± 24 min). A total of 83.3% of the patients underwent coronary angiography and in 93.3% of them acute coronary syndrome (ACS) was confirmed to be the cause of shock-resistant VF. Three patients did not undergo coronary angiography, as the on-call cardiologists had pronounced them dead prior to the procedure. In all the patients with ACS as the confirmed cause of cardiac arrest, PCI was attempted, with 50.0% of these patients still undergoing mechanical CPR. A total of 60% of the patients survived the procedure and were transferred to an intensive care unit. Of these, five patients survived to discharge, accounting for 27.8% of the initially enrolled patients, and 33.3% of the patients who had undergone coronary angiography. All the discharged patients were in good neurological condition. Four of these patients had undergone PCI after ROSC, and in one patient PCI was performed during CPR with mechanical support. 

Procedural data are shown in Table 2. Analysis of the group of patients who survived at least 24 h after the procedure (50% of the patients) revealed that the duration of cardiac arrest was significantly shorter and averaged 18.8 ± 17 min vs 74.9 ± 45 min in those who did not survive the procedure (*p* = 0.013) and that ROSC during transport to the hospital occurred in a larger percentage of these patients (100% vs. 37.5%, *p* = 0.038), so that the AP device was less frequently used during PCI in these patients (11.1% vs. 75.0%, *p* = 0.027). Analysis of the subgroup of patients who survived until being discharged from hospital revealed that the mean duration of CPR was considerably shorter in those who survived with a good outcome (20 ± 22 min vs. 58 ± 45 min). The difference was not, however, significant, although a significant trend was noted (*p* = 0.064). No significant relationship was observed between long-term survival and the culprit coronary artery. 

## 4. Discussion

We prospectively evaluated the use of mechanical chest compression in patients with shock-resistant VF. As the eligibility criterion, we adopted the necessity to deliver at least three consecutive shocks. None of the previous studies had focused on such a patient group selected in this manner. In addition, PCI is still rarely performed during cardiac arrest [7], which in our study was the case in 38.8% of the entire study population. In previous studies of various devices for mechanical chest compression, the number of patients with shockable rhythms (VF/VT) ranged from 5 to 6 [8]. Previously published studies focused on comparing mechanical versus manual CRP [9]. Several reports have, however, been published on shock-resistant VF where the efficacy of the following drugs was assessed: amiodarone and lidocaine [3], and nifekalant [10]. In the nifekalant studies, both Harayama et al. and Shiga et al. defined resistant VF as the necessity to deliver at least two shocks [10]. In another study, resistant VF was defined as VF persisting after one shock [11]. None of the previous studies reported on mechanical chest compression and direct transport to a catheterization laboratory in patients with resistant VF defined as persisting after a minimum of three consecutive shocks. However, irrespective of the study group selected, high-quality chest compression during cardiac arrest is a key determinant of the outcome: patient survival [12]. Many studies have reported good ROSC rates when using devices for chest compression. Other studies, however, have not demonstrated significant differences in favor of mechanical chest compression [13]. The American Heart Association (AHA) and the European Resuscitation Council (ERC) stress the importance of improving the quality of chest compressions as the key factor contributing to ROSC—something that is difficult to achieve during patient transport to the hospital or even more so during a PCI procedure. Therefore, the ERC highly recommends the use of devices for mechanical chest compression, especially during PCI procedures [14]. This was an exceptional advantage in our study, as in 50.0% of the patients who underwent PCI, the procedure was continued during CPR. Various devices for automatic CPR are used on the market: AutoPulse, which we used in our study, and LUCAS and LifeStat. We used the AutoPulse platform in our study as all members of the ambulance team in the area of interest were equipped with it. Consistent quality of compressions is the advantage shared by all the devices for mechanical chest compression [15]. However, both AutoPulse and LUCAS cause some discomfort during radiological imaging in the anteroposterior dimension. In the case of AutoPulse, the device’s electronics is the problem in anteroposterior views, while in the case of LUCAS, the problem is caused by the large piston which considerably interferes with movements of the angiography system C-arm. Nevertheless, in the case of both devices it is possible to obtain the basic cranial and caudal views, which are frequently used during PCI. Each device that delivers high-quality compressions greatly shortens the time to reaching the hospital, as it allows the responders to carry out other important tasks, such as performing endotracheal intubation, administering drugs, etc. It is also invaluable that such devices allow the crew to move through narrow corridors or walk down several flights of stairs while continuing to administer CPR [9]. We did not evaluate the duration of transport to the hospital during mechanical chest compression and without mechanical chest compression, as this was beyond the scope of our study. However, the mean duration of treatment (60 min) provided by the ambulance team until patient handover to the catheterization laboratory, assuming that the ambulance crew consisted of two members only, seems acceptable in comparison with other studies in which longer transport lag times were reported [16]. On the other hand, in another group of patients, the transport lag time depended on the distance to the catheterization laboratory and the patient’s condition and ranged from 38 to 115 min [17]. In those studies, however, the duration of transport to the catheterization laboratory was not reported for the subgroups of patients with ongoing cardiac arrest. Also, according to the 2017 European Society of Cardiology (ESC) guidelines on the management of acute coronary syndromes with ST-segment elevation, the recommended time to PCI should be less than 120 min, and in patients with ST-segment elevation after cardiac arrest, immediate PCI should be the strategy of choice [6]. In our project, we assumed direct transport to the catheterization laboratory in cases of patients with resistant VF and a coronary angiography attempt with a possible subsequent PCI, irrespective of whether ST-segment elevation was identified after ROSC or not. The course of action we adopted in our study proved to be right, as among the patients who were considered alive at the catheterization laboratory and proceeded to coronary angiography, 93.3% were found to have ACS as the cause of cardiac arrest, as confirmed by the visualization of a coronary artery lesion. In all these patients, PCI was attempted and in half of the cases successful, which allowed a total of five patients to survive until discharge from hospital in a good neurological condition. Also, in other described registries, in patients after cardiac arrest and ROSC, routine coronary angiography and, possibly, revascularization considerably increased the chance of survival [18]. In these studies, however, it was not immediate, as in our analysis. This course of action seems even more important in patients with shock-resistant VF, in whom the chances of confirming ACS are even higher, as confirmed by our study. One important finding in our study is that even though in most cases resuscitation was prolonged, and transport took an average of one hour, more than half of the patients (55.5%) were handed over to the catheterization laboratory while still in cardiac arrest. Only one of these patients survived to discharge. On the other hand, in the group of patients handed over after ROSC, half of the patients survived to discharge. None of the variables affected ROSC in our study. A significant trend towards ROSC was detected in patients with a shorter duration of CPR only, a finding consistent with previous studies [19]. In addition to the above, effective BLS is another known key element that increases the chances of ROSC [20]. BLS was an eligibility criterion in our study and was provided to all our patients, but it was not possible to assess its correctness. Increased chances of ROSC have also been reported with lidocaine vs amiodarone in children with cardiac arrest in shockable rhythms [21]. Also in our study, amiodarone did not increase the chance of ROSC. Of all the laboratory parameters included in our analysis, only troponin was significantly higher in the group of patients who did not survive to discharge. It may, therefore, be concluded that in these patients, ACS might have lasted slightly longer than in the subgroup of patients who survived to discharge and experienced more extensive myocardial necrosis. Obviously, elevated troponin levels could also have resulted from prolonged CPR in these patients. However, in other studies that evaluated troponin levels after SCA, this parameter was significantly higher in patients with coronary artery closure, prolonged CPR, heart failure and renal failure [22]. In the study by Røsjø et al. cited above, troponin levels were also higher in non-survivals versus survivals, a finding consistent with our study [22]. When analyzing survival of patients in this study, it is difficult to compare this population with other studies, as previous research did not focus on patients with resistant VF. The survival rates reported in this study, at 27.8% in the entire group of patients transported to the catheterization laboratory and at 33.3% in patients undergoing coronary angiography, are, however, comparable with those reported for the large group of patients in the registry of Brennan et al., where the mortality rate of patients who suffered cardiac arrest during transport to the catheterization laboratory was 64% [23]. Survival in the OHCA population, on the other hand, is much lower than that in our study and ranges from 5% to 8% [24]. However, these data do not include all the underlying mechanisms of cardiac arrest. It is widely known that PEA and asystole are associated with a much worse prognosis, and in one study a good neurological outcome was reported in 7% and 2.7% of cases, respectively, compared with 36.4% in VF [25]. Other studies, on the other hand, reported up to 100% mortality in non-shockable rhythms [14,26]. These results are, however, quite similar if these survival results are compared to a study conducted in groups of patients undergoing PCI during cardiac arrest. In this study, survival was reported at 25% [8]. This study assessed post-PCI survival in patients who had arrested at the catheterization laboratory, which considerably shortens the time to provision of professional care and, even more so, significantly increases the possibility of opening a coronary vessel. In this light, one should favorably view the survival result of nearly 30% in our study, where the mean duration of transport was 60 min (median, 69 min) and the mean duration of CPR was 48 min (median, 36.5 min).

Limitations of the study: One of the limitations was the fact that the patients handed over to the catheterization laboratory had initially been triaged by the ambulance team at the scene, following a telephone conversation with the on-call cardiologist. No data are available on the patients who were not considered eligible for the study and most of whom died at the scene. It is, therefore, difficult to evaluate the correctness of each decision to exclude those patients from the study. Also, no data are available on the effectiveness of BLS, which is one of the links in the survival chain and is of key importance to patient survival. 

## 5. Conclusions

The use of AutoPulse during transport to the catheterization laboratory in the analyzed group of patients with shock-resistant VF allowed better survival ratios to be achieved than those previously reported in the literature among patients receiving CPR at the scene of the incident. However, the group was quite small and, therefore, theresults cannot be generalized. The shortest duration of CPR and ROSC during transport to the hospital are two key factors in patient survival after the procedure. AP is helpful in conducting CPR and makes it possible to reduce the time from cardiac arrest to the arrival at a catheterization laboratory. Resistant VF suggests a very high likelihood of a coronary component to the cardiac arrest; the patient should, therefore, be directly transported to a catheterization laboratory, and PCI should be attempted despite the frequently still ongoing arrest, as this considerably increases the patient’s chance of survival. The price of AutoPulse is affordable, so it is feasible to place AP in all ambulances. The Autopulse is important in refractory cardiac agrest. It could help to guarantee adequate blood circulation to place an ECMO and aortic counterpulsation. In the case of withdrawal of therapy for persistent refractory cardiac arrest, the previous use of AutoPulse could help the DCD (donation after circulatory death) procedure [27,28].

## Figures and Tables

**Table 1 ijerph-19-02557-t001:** Study group characteristics.

Characteristic Whole Group	Median ± IQR	Dead Median ± IQR	Alive Median ± IQR	*p*-Value
Age (years)	62.5 ± 4	62 ± 4	63 ± 17	0.427
Number of shocksdelivered	4 ± 1	4 ± 1	3 ± 1	0.633
Responsetime (min)	4 ± 2	5 ± 3.5	4 ± 1	0.194
Time of transport to the catheterization laboratory [min]	69 ± 24	59 ± 20	73.5 ± 24	0.342
Duration of CPR (min)	36.5 ± 71.5	44 ± 91.5	11 ± 14	0.063
Adrenalinedose (mg)	4.5 ± 3	4 ± 3	5 ± 2	0.129
Troponin (pg/mL)	151 ± 93	268 ± 753	60 ± 84	0.030
CK-MB (U/L)	129.5 ± 93	141 ± 181	116.5 ± 75	0.761
CRP (mg/dL)	1.4 ± 3	2.3 ± 2.7	1.0 ± 0.5	0.177
Glucose (mg/dL)	292 ± 231	300 ± 176	206 ± 240	0.431
Sodium (mmol/L)	138 ± 7	137 ± 9	139 ± 5	1.0
Potassium (mmol/L)	4.2 ± 1.6	4.9 ± 1.8	3.6 ± 1.1	0.202
Chloride (mmol/L)	106 ± 9	104.5 ± 9	107.5 ± 18	0.609
Urea (mg/dL)	47 ± 24	47 ± 13	49 ± 24	0.857
RBC (mln/mL)	4.66 ± 0.6	4.35 ± 0.5	4.84 ± 0.7	0.527
WBC (thous/mL)	13.8 ± 5	13.8 ± 4.6	13.3 ± 13.4	0.927
PLT (thous/mL)	176 ± 44	160 ± 64	181 ± 43	0.412
HGB (mmol/L)	9.1 ± 1.4	9.1 ± 2.6	9.3 ± 1.3	0.516
Creatinine (mg/dL)	1.2 ± 0.68	1.61 ± 0.72	1.03 ± 0.15	0.431
GFR (mL/min/1.73m^2^)	56 ± 27	50 ± 16	64 ± 34	0.476

CPR—cardiopulmonary resuscitation; CK–MB—creatine kinase MB isoenzyme; CRP—C-reactive protein; RBC—red blood cells; WBC—white blood cells; PLT—platelets; HGB—haemoglobin; GFR—glomerular filtration rate.

**Table 2 ijerph-19-02557-t002:** Procedural data.

	*n* (%)	Dead *n* (%)	Alive *n* (%)
Coronary angiography	15 (83.3)	10 (66.7)	5 (33.3)
ACS confirmed	14 (77.8)	10 (71.4)	4 (28.6)
PCI	14 (77.8)	10 (71.4)	4 (28.6)
PCI continued during resuscitation with AutoPulse™	7	6 (85.7)	1 (14.3)
Culprit artery:			
LMCA	2 (14.2)	2 (100)	0 (0)
LAD	6 (42.9)	4 (66.7)	2 (33.3)
LCx	0 (0)		
RCA	6 (42.9)	4 (66.7)	2 (33.3)
Handover during SCA	10 (55.5)	9 (90.0%)	1 (10.0%)
Handover after ROSC	8 (44.5)	4 (50.0%)	4 (50.0%)
ROSC	11 (64.7)	6 (54.6%)	5 (45.4%)

ACS—acute coronary syndrome; LMCA—left main coronary artery; LAD—left artery descending coronary artery; LCx—left circumflex coronary artery; PCI—percutaneous coronary intervention; RCA—right coronary artery; ROSC—return of spontaneous circulation; SCA—sudden cardiac arrest.

## Data Availability

Data supporting the reported results can be accessed by corresponding with the authors.

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
