# Peer review of "Efficacy of AutoPulse for Mechanical Chest Compression in Patients with Shock-Resistant Ventricular Fibrillation"

_ijerph, 2022, doi:10.3390/ijerph19052557_

Round 1

Reviewer 1 Report

Submitted manuscript on the effectiveness of the AutoPulse (AP) system for mechanical chest compression in patients with refractory VF transported directly to the catheterization laboratory during resuscitation, represents an original approach to an ever-pressing topic. The use of mechanical chest compression devices continues to increase. Emergency medical teams and hospital emergency departments use them frequently.

The authors take on an important topic that finds its justification in the ERC Resuscitation Guidelines 2021. This topic should be developed and taken up nationwide to know how often we use these devices.

My comments and questions to the Authors:

  1. Please explain in the methodology why only the effectiveness of the AutoPuls was discussed when there are other devices available on the market.
  2. Please indicate from which clinical centers the data was collected.
  3. Please indicate whether the researchers (Authors) participated directly in the study e.g. if they were members of the emergency teams or interventional cardiology laboratory, or were passive observers.
  4. Was the AutoPulse model 100 AutoPulse® resuscitation system (also known as the AutoPulse)? Or another? This should be specified in the introduction, and briefly about its design and main component?
  5. It is important to emphasize the patient/operating parameters of the AutoPulse system, i.e.  "The AutoPulse System is intended for adults weighing 136 kg or less,with a chest circumference of 76 to 130 cm and a chest width of 25 to 38 cm. "
  6. Table "Table 2. Procedural data." not very clear. Isn't the date missing?
  7. Do the authors know in which Compression Mode (user selectable) the selected device model operated? Could it affect the results of the study?
  8. Conclusions should correspond more to the topic of the paper, the purpose of the paper and the results.

Reviewer 2 Report

Congratulations to the authors. Thank you for the opportunity to review this manuscript. Some considerations for improvement this paper are in the following comments.

Line 43. “Mechanical chest compression devices are useful in cases of pro- 43 longed cardiac arrest” Can you provide some reference for this statement?

Line 49. The study was a multicentre 49 prospective observational clinical trial. This has already been said in the beginning of methods, which is the right place. delete it from here

Line 53. Patients and Methods. Delete “patients”, this is implicit in the method

Line 76.  “No possibility of rapid evacuation of the patient from the scene”. Can you provide a time interval?

Line 116. Ethics. This section should go before the statistical analysis

Line 192. “During the one-year follow-up period, a total of call-outs for patients with SCA were reported in the study population of thousand people inhabiting the area of interest”. This sentence was already written in methods, it should be removed from results since it is not a result of your study.

Conclusions.

“The use of AutoPulse during transport to the catheterisation laboratory in patients with shock-resistant VF allows better survival rates to be achieved than those previously reported in the literature among patients receiving CPR at the scene of the incident”

I think this paragraph should be reformulated. With the number of cases analyzed, I do not believe that its results can be generalized. Perhaps clarify that for the analyzed sample higher survival ratios were found.

Reviewer 3 Report

This is an interesting paper, in my point of view the topic of autopulse is the next step in the future of BLS-D. 

The limit of the paper is represented by the sample size, but the methodology is well described and results clearly expressed.

I have only some suggestions to recommend to the authors: cost of autopulse should be explicated: Is It feasible to place in all ambulances?

How many aortic counterpulsation were placed in case of the refractory arrest?

In the end, the autopulse offers another chance in case of refractory cardiac arrest: in fact, it could help to guarantee blood circulation adequate to place an ECMO and aortic counterpulsation. In case of withdrawal of therapy for persistent refractory cardiac arrest the previous use of autopulse could help the DCD procedure. [Please I suggest and recommend discussing the topic, you can choose the article you prefer] Manara AR, Murphy PG and O'Callaghan G. Donation after circulatory death. Br J Anaesth 2012; 108 Suppl 1: i108-121. DOI: 10.1093/bja/aer357  or  Baroni S, Melegari G, Brugioni L, et al. First experiences of hemoadsorption in donation after circulatory death. Clin Transplant 2020; 34: e13874. 20200427. DOI: 10.1111/ctr.13874.

Round 2

Reviewer 1 Report

Accept in present form